# The Level of Remnant Cholesterol and Implications for Lipid-Lowering Strategy in Hospitalized Patients with Acute Coronary Syndrome in China: Findings from the Improving Care for Cardiovascular Disease in China—Acute Coronary Syndrome Project

**DOI:** 10.3390/metabo12100898

**Published:** 2022-09-24

**Authors:** Na Yang, Miao Wang, Jing Liu, Jun Liu, Yongchen Hao, Dong Zhao

**Affiliations:** Department of Epidemiology, Beijing Anzhen Hospital, Capital Medical University, Beijing Institute of Heart, Lung and Blood Vessel Diseases, Beijing 100029, China

**Keywords:** acute coronary syndrome, remnant cholesterol, low-density lipoprotein cholesterol, non-high-density lipoprotein cholesterol

## Abstract

Elevated remnant cholesterol is associated with an increased risk of atherosclerotic cardiovascular disease (ASCVD). We aimed to evaluate the concentrations and general distribution of remnant cholesterol at admission in patients hospitalized for acute coronary syndrome (ACS), and those in patients who reached the low-density lipoprotein cholesterol (LDL-C) target or non-high-density lipoprotein cholesterol (non-HDL-C) target. Patients with ACS who were enrolled in the Improving Care for Cardiovascular Disease in China—ACS project from 2014 to 2019 were included. Elevated remnant cholesterol concentrations were defined as ≥1.0 mmol/L. Among 94,869 patients, the median (interquartile range) remnant cholesterol concentration at admission was 0.6 mmol/L (0.4–0.9 mmol/L) and 19.2% had elevated remnant cholesterol concentrations. Among patients with LDL-C concentrations < 1.4 mmol/L, 24.4% had elevated remnant cholesterol concentrations, while the proportion was 13.3% among patients with LDL-C concentrations between 1.4 and 1.7 mmol/L. Among patients with non-HDL-C concentrations < 2.6 mmol/L, 2.9% had elevated remnant cholesterol concentrations but 79.6% had LDL-C concentrations ≥ 1.4 mmol/L. Even among patients with LDL-C < 1.4 mmol/L and non-HDL-C < 2.6 mmol/L, 10.9% had elevated remnant cholesterol. In conclusion, one fifth of patients with ACS have elevated remnant cholesterol concentrations at admission. Elevated remnant cholesterol concentrations are present in patients with LDL-C or/and non-HDL-C concentrations within the target, which represents an unmet need to add remnant cholesterol as a target for the secondary prevention of ASCVD.

## 1. Introduction

Cardiovascular disease (CVD), mainly atherosclerotic cardiovascular disease (ASCVD), is the leading cause of death worldwide [1]. Low-density lipoprotein cholesterol (LDL-C) is the most well-established risk factor for ASCVD [2]. LDL-C has been recommended as the primary target for lipid-lowering therapies for the secondary prevention of ASCVD in almost all guidelines [3,4,5]. With the advent of LDL-C-lowering therapies beyond statins, LDL-C concentrations can be reduced to lower levels. However, the event rates of recurrent ASCVD remained 9.5% in the ODYSSEY OUTCOMES study at a median follow-up of 2.8 years and 9.8% in the FOURIER study at a median follow-up of 2.2 years, although LDL-C concentrations were reduced to 1.8 mmol/L or lower [6,7]. Cholesterol in triglyceride (TG)-rich lipoproteins (TRLs), including the cholesterol contained in very-low-density lipoproteins and intermediate-density lipoproteins in the fasting state, and also chylomicron remnants in the non-fasting state, are collectively named remnant cholesterol. Accumulating evidence has suggested that remnant cholesterol is an important source of the residual risk of ASCVD [8,9,10]. Findings from Mendelian randomization studies have supported a causal relationship between remnant cholesterol and ischemic heart disease [11,12].

Non-high-density lipoprotein cholesterol (non-HDL-C), which includes LDL-C and all the other atherogenic cholesterols, is recommended as a secondary lipid-lowering target after achieving the LDL-C target for the prevention of ASCVD in several guidelines. Additionally, the 2013 position paper from the International Atherosclerosis Society and the 2015 National Lipid Association recommendations for the management of dyslipidemia recommend non-HDL-C as a primary target [13,14]. Patients with ACS are categorized as being at very high risk or extreme risk of ASCVD on the basis of their clinical profile. The guideline recommends a primary target of LDL-C concentrations <1.8 mmol/L for patients at very high risk and 1.4 mmol/L for those at extreme risk. Additionally, a secondary target of non-HDL-C concentrations < 2.6 mmol/L is recommended for patients at very high risk and an even lower concentration is recommended for those at extreme risk. In patients with LDL-C and/or non-HDL-C concentrations within the target, information on the remnant cholesterol burden is lacking. Emerging therapies targeting TRLs and their remnants have been under investigation [8,15]. Therefore, identifying the concentrations and distribution of remnant cholesterol in patients with ASCVD, as well as in subgroups with LDL-C or/and non-HDL-C concentrations within the target, is important. This information is essential to identify the demand for a reduction in remnant cholesterol and may have important implications for lipid-lowering strategies. However, few studies have addressed this issue. Therefore, this study aimed to determine the concentrations and distribution of remnant cholesterol in patients with ACS and in subgroups of ACS patients with LDL-C and/or non-HDL-C concentrations within the target.

## 2. Materials and Methods

### 2.1. Study Design

The study used data from the Improving Care for Cardiovascular Disease in China (CCC)—ACS project, which is a collaborative quality improvement project of the American Heart Association and the Chinese Society of Cardiology. Details regarding the study design and methodology have been published elsewhere [16]. Briefly, the project was initiated in November 2014. Hospitals were recruited across mainland China, stratified by geographic–economic levels. A total of 159 tertiary hospitals and 82 secondary hospitals were recruited. Each month, participating tertiary hospitals reported the first 20–30 eligible patients, and each secondary hospital reported the first 10–20 eligible patients with a principal discharge diagnosis of ACS to the CCC-ACS project in a consecutive manner. The project used a web-based data collection platform (Oracle Clinical Remote Data Capture; Oracle Corporation, Redwood City, CA, USA) to collect data regarding patients’ characteristics, symptoms on arrival, laboratory testing, in-hospital management and outcomes, and discharge treatment. All of the information was abstracted from patients’ medical records by trained abstractors. Multiple approaches, such as real-time online data checks, regular on-site quality audits, and training workshops, were adopted to ensure the accuracy and completeness of the data.

### 2.2. Study Population

A total of 113,650 patients with a principal discharge diagnosis of ACS were recruited to the CCC-ACS project from November 2014 to December 2019. We excluded 18,754 patients with missing values of total cholesterol (TC), LDL-C, or HDL-C. Finally, 94,869 patients were included in the present study.

### 2.3. Measurement of Lipid Profiles

The lipid profile was measured using standard hospital assays. The directly measured method was recommended as the industry standard for routine measurement of LDL-C in China. Therefore, nearly all the hospitals reported directly measured LDL-C concentrations. Routinely, blood samples were collected after overnight fasting. Remnant cholesterol concentrations were calculated as TC minus HDL-C minus LDL-C concentrations [12,17].

### 2.4. Definition of Variables

There is no consensus on the optimal remnant cholesterol concentrations in patients with ACS. In the present study, elevated remnant cholesterol concentrations were defined as ≥1.0 mmol/L. This definition was used in accordance with a previous study, which showed a significantly increased risk of recurrent ASCVD associated with remnant cholesterol concentrations ≥1.0 mmol/L [18]. LDL-C concentrations were categorized as clinically relevant groups of <1.4 mmol/L, ≥1.4 and <1.8 mmol/L, ≥1.8 and <2.6 mmol/L, and ≥2.6 mmol/L. Non-HDL-C concentrations were divided into <2.6 mmol/L, ≥2.6 and <3.4 mmol/L, ≥3.4 and <4.1 mmol/L, and ≥4.1 mmol/L. Similarly, TG concentrations were categorized as <1.7 mmol/L, ≥1.7 and <2.3 mmol/L, ≥2.3 and <5.7 mmol/L, and ≥5.7 mmol/L. Smoking was defined as smoking within the preceding 1 year. Hypertension was defined as having a history of hypertension, receiving antihypertensive drugs before admission, or having a systolic blood pressure ≥140 mmHg or diastolic blood pressure ≥90 mmHg at admission. Diabetes mellitus was defined as having a history of diabetes mellitus, receiving oral hypoglycemic drugs or insulin therapy before admission, or having fasting blood glucose concentrations ≥ 7.0 mmol/L or hemoglobin A1c values ≥ 6.5%. A history of CVD was defined as a history of myocardial infarction, peripheral vascular disease, stroke/transit ischemic attack, percutaneous coronary intervention, or coronary artery bypass grafting prior to the hospitalization. Prehospital statin use was defined as receiving statins in the past 2 weeks before admission.

### 2.5. Statistical Analysis

Continuous variables are presented as the mean ± standard deviation or the median (interquartile range (IQR)), as appropriate. Categorical variables are presented as the number (%). The t-test, chi-squared test, or Mantel–Haenszel chi-square test was used to compare the differences in the patients’ characteristics between men and women, as appropriate. A sequential regression multiple imputation method was used to impute the missing values of variables with a missing rate of <15%. The imputation was performed using IVEware software version 0.2 (Survey Research Center, University of Michigan, Ann Arbor, MI, USA). All statistical analyses were performed using SAS software version 9.4 (SAS Institute, Cary, NC, USA). A two-sided *p* value of <0.05 was considered significant.

## 3. Results

### 3.1. Patients’ Characteristics

Among 94,896 patients with ACS, 42.5% had ST-segment elevation myocardial infarction (STEMI) (Table 1). The mean age was 63.3 ± 12.4 years and 26.6% were women. A total of 67.1% of patients had hypertension, 43.4% had diabetes, and 27.4% had a history of CVD. A total of 17.6% were on statin treatment at admission. The mean concentrations of LDL-C, non-HDL-C, and HDL-C of the patients at admission were 2.7, 3.4, and 1.1 mmol/L, respectively. The median TG concentration at admission was 1.4 mmol/L. Only 6.4% of patients with ACS had LDL-C concentrations < 1.4 mmol/L, 16.2% had LDL-C concentrations < 1.8 mmol/L, 23.3% had non-HDL-C concentrations < 2.6 mmol/L, 5.4% had LDL-C concentrations < 1.4 mmol/L and non-HDL-C concentrations < 2.6 mmol/L, and 8.4% had LDL-C concentrations of 1.4–1.7 mmol/L and non-HDL-C concentrations <2.6 mmol/L.

### 3.2. Distribution of Remnant Cholesterol

The distribution of remnant cholesterol in patients hospitalized for ACS was right-skewed with a tail towards higher concentrations (*p* < 0.01, Figure 1). The mean remnant cholesterol concentration was 0.7 ± 0.6 mmol/L. The median (IQR) remnant cholesterol concentration was 0.6 mmol/L (0.4, 0.9 mmol/L). The 5% and 95% quantiles of remnant cholesterol concentrations were 0.2 and 1.6 mmol/L, respectively. A total of 19.2% of patients had elevated remnant cholesterol concentrations (≥1.0 mmol/L).

### 3.3. Proportion of Remnant Cholesterol in TC

The proportion of remnant cholesterol, HDL-C, and LDL-C among the TC was 16.0%, 23.9%, and 60.1%, respectively, which indicated that LDL-C accounted for the highest proportion (Figure 2). When patients were divided into four subgroups based on TG concentrations, the proportion of patients in each group was 63.5%, 17.2%, 17.6%, and 1.8%, respectively. The proportion of LDL-C in TC was 60.8% in patients with TG concentrations < 1.7 mmol/L, which decreased to 43.7% in patients with TG concentrations ≥ 5.7 mmol/L. In contrast, the proportion of remnant cholesterol in TC was 13.0% in patients with TG concentrations < 1.7 mmol/L and increased to 40.1% in patients with TG concentrations ≥ 5.7 mmol/L.

### 3.4. Remnant Cholesterol Concentrations in Patients with Different LDL-C or Non-HDL-C Concentrations

When the patients were divided into four subgroups based on LDL-C concentrations, the fourth quintile of remnant cholesterol concentrations in patients with LDL-C concentrations < 1.4, 1.4–1.7, 1.8–2.5, and ≥2.6 mmol/L was 1.2, 0.8, 0.9, and 1.0 mmol/L, respectively, which was the highest among patients with LDL-C concentrations < 1.4 mmol/L (Appendix A). These results indicated that, among patients with ACS, even in those with LDL-C concentrations within a stricter treatment target, there was a higher proportion of patients whose remnant cholesterol concentrations were at relatively higher levels. The proportions of patients with elevated remnant cholesterol concentrations were 24.4%, 13.3%,16.0%, and 21.7% in those with LDL-C concentrations <1.4, 1.4–1.7, 1.8–2.5, and ≥2.6 mmol/L, respectively.

When the patients were divided into four subgroups based on non-HDL-C concentrations, the fourth quintile of remnant cholesterol concentrations was 0.6, 0.8, 1.0, and 1.4 mmol/L in patients with non-HDL-C concentrations < 2.6 mmol/L, 2.6–3.3 mmol/L, 3.4–4.1 mmol/L, and ≥4.1 mmol/L, respectively. The proportions of patients with elevated remnant cholesterol concentrations were 2.9%, 9.6%, 21.7%, and 44.3% in these four subgroups, respectively (Appendix A). The proportion of patients with LDL-C concentrations ≥ 1.8 mmol/L was 41.0% among patients who had met the treatment target for non-HDL-C (<2.6 mmol/L, Appendix A). Among patients in the other three groups, the proportion of patients with LDL-C concentrations ≥1.8 mmol/L was more than double the proportion in patients with non-HDL-C concentrations <2.6 mmol/L.

### 3.5. Remnant Cholesterol Concentrations in Patients with LDL-C and/or Non-HDL-C Concentrations within the Treatment Target

The quintile cut-offs and median remnant cholesterol concentrations in patients whose LDL-C or non-HDL-C concentrations met the treatment target at admission are shown in Figure 3. The proportion of patients with elevated remnant cholesterol concentrations was 24.4% and 13.3% in patients with LDL-C concentrations < 1.4 mmol/L and 1.4–1.7 mmol/L, respectively (Figure 3a,b). The proportion of patients with elevated remnant cholesterol concentrations was only 2.9% in patients whose non-HDL-C concentrations achieved the target concentration (<2.6 mmol/L) at admission (Figure 3c). However, the proportion of patients with LDL-C concentrations ≥1.8 mmol/L was 41.0% in patients with non-HDL-C concentrations within the target, and the proportion of patients with LDL-C concentrations ≥ 1.4 mmol/L was 79.6% (Figure 3d). Among patients with LDL-C concentrations < 1.8 mmol/L and non-HDL-C concentrations < 2.6 mmol/L, 5.0% had elevated remnant cholesterol concentrations. Among patients with LDL-C concentrations < 1.4 mmol/L and non-HDL-C concentrations < 2.6 mmol/L, 10.9% of patients still had elevated remnant cholesterol concentrations (Figure 3e). Among patients with LDL-C concentrations of 1.4–1.7 mmol/L and non-HDL-C concentrations < 2.6 mmol/L, 1.2% of patients had elevated remnant cholesterol concentrations (Figure 3f).

Additional analysis showed the proportion of remnant cholesterol in TC increased with decreasing concentrations of LDL-C (Appendix A). The proportion of remnant cholesterol in TC was 29.1% among patients with LDL-C concentrations < 1.4 mmol/L. The proportion of remnant cholesterol in TC increased with increasing concentrations of non-HDL-C, with the proportion of 18.1% in patients with non-HDL-C ≥ 4.1 mmol/L. In addition, the proportion was 20.4% even among patients with LDL-C and non-HDL-C concentrations both within the targets.

In the subgroup analysis (Table 2), in patients with LDL-C concentrations < 1.4 mmol/L at admission, 27.0% had elevated remnant cholesterol concentrations in those with STEMI, and this was higher than that in those with NSTE-ACS (22.4%, χ^2^ = 17.256, *p* < 0.001). In patients with LDL-C concentrations < 1.4 mmol/L and non-HDL-C concentrations <2.6 mmol/L, 11.4% had elevated remnant cholesterol concentrations in those with STEMI and this rate was similar to that in those with NSTE-ACS (10.5%, χ^2^ = 0.935, *p* = 0.334). In patients with LDL-C concentrations < 1.4 mmol/L, the proportion of patients with elevated remnant cholesterol concentrations was higher in those without a history of CVD than in those with a history of CVD (29.5% vs. 17.5%, χ^2^ = 114.373, *p* < 0.001). 

## 4. Discussion

Using a nationwide real-world registry, we evaluated the remnant cholesterol burden in patients with ACS and in patients with ACS and LDL-C or/and non-HDL-C concentrations within the guideline-recommended target. The main findings were as follows. First, the distribution of remnant cholesterol at admission in patients with ACS was right-skewed, with great variation among the patients, and nearly 20% had elevated remnant cholesterol concentrations. Second, with increasing TG concentrations, the proportion of remnant cholesterol in TC increased, while the proportion of LDL-C decreased. Third, in patients with LDL-C concentrations within the target, a considerable proportion of patients had elevated remnant cholesterol concentrations. In patients with non-HDL-C concentrations within the target, although less patients had elevated remnant cholesterol, a high proportion of the patients did not reach the LDL-C target.

A few previous studies reported similar concentrations and distributions of remnant cholesterol in patients with ASCVD as those in our study. Shao et al. found that the median fasting remnant cholesterol concentration was 0.58 mmol/L (IQR: 0.43–0.79 mmol/L) in Chinese patients with ACS undergoing percutaneous coronary intervention [19]. Among patients with a history of myocardial infarction or ischemic stroke who were enrolled in the Copenhagen General Population Study, a similar distribution to that in our study, with a tail toward higher concentrations, was observed, and 26.6% of the population had non-fasting remnant cholesterol concentrations ≥ 1 mmol/L [18]. The proportion was higher in the Copenhagen General Population Study than in our study because a non-fasting blood sample was adopted. Previous studies have shown that remnant cholesterol concentrations ≥1 mmol/L are associated with a 30% higher risk of all-cause mortality in patients with ischemic heart disease [20]. In addition, the Copenhagen General Population Study has shown a dose–response relationship between remnant cholesterol and recurrent major cardiovascular events (MACE) in patients with ASCVD There was a 48% and 79% higher risk of MACE in patients with remnant cholesterol concentrations of 1–1.49 mmol/L and ≥1.5 mmol/L, respectively, than in those with remnant cholesterol concentrations < 0.5 mmol/L [18]. Post hoc analysis from the Treating to New Targets trial showed that, in patients with CVD, a reduction in remnant cholesterol concentrations with statins was associated with a lower risk of MACE independent of LDL-C concentrations [21]. Thus, data from our study showed remnant cholesterol-lowering therapies may be indicated in nearly 20% of patients with ACS whose remnant cholesterol concentrations were ≥1 mmol/L.

In this study, the proportion of remnant cholesterol in TC increased, while the proportion of LDL-C decreased with increasing concentrations of TG. Our finding is consistent with that from the Copenhagen Ischemic Heart Disease Study, although this previous study measured TG and remnant cholesterol in the non-fasting state [20]. TG are a marker of TRLs, although TG per se is not likely to cause atherosclerosis [22,23]. Remnant cholesterol was the cholesterol content contained in all TRLs. Therefore, remnant cholesterol was positively correlated with TG. In patients with extremely high TG concentrations, the proportion of remnant cholesterol in TC was >40%, which indicated that patients with ACS and hyperglyceridemia might be a target population for reducing remnant cholesterol

Even in ACS patients with LDL-C concentrations < 1.4 mmol/L, nearly one quarter had elevated remnant cholesterol concentrations, and the proportion was 13.3% in patients with LDL-C concentrations between 1.4 and 1.7 mmol/L. Elevated remnant cholesterol concentrations may partly explain the occurrence of ACS, despite low LDL-C concentrations in these patients. Similar to LDL, remnant lipoproteins can penetrate the arterial wall, leading to the accumulation of remnant cholesterol in the arterial intima, formation of atherosclerotic plaques, and, finally, ASCVD [24,25]. Additionally, TRLs may cause atherosclerosis by other pathways, such as an enhanced inflammatory response [26]. A Mendelian randomization study also showed that elevated remnant cholesterol concentrations were causally associated with low-grade inflammation [12]. Therefore, in patients with LDL-C concentrations within the target, there is still an unmet need to treat elevated remnant cholesterol concentrations. LDL-C alone as a lipid-lowering target may not be sufficient for the secondary prevention of ASCVD. Remnant cholesterol may be a secondary target for lipid-lowering therapies after achieving the LDL-C target.

In our study, among patients with non-HDL-C concentrations within the target, only 3% of them had elevated remnant cholesterol concentrations. However, more than 40% of the patients had LDL-C concentrations above the target, and nearly 80% had LDL-C concentrations ≥1.4 mmol/L. These findings indicated that the risk related to elevated LDL-C concentrations could not be fully reduced if non-HDL-C alone was adopted as a lipid-lowering target. In patients with both LDL-C concentrations (<1.8 mmol/L) and non-HDL-C concentrations (<2.6 mmol/L) within the target, we still observed elevated remnant cholesterol concentrations in 5.0% of the patients. This proportion was even higher than 10% in patients with LDL-C concentrations ≤ 1.4 mmol/L and non-HDL-C concentrations within the target. Therefore, the risk related to elevated remnant cholesterol concentrations cannot be fully reduced, even in patients with both LDL-C and non-HDL-C concentrations within the target. Taken together, our findings suggest that LDL-C and remnant cholesterol as a separate lipid-lowering target, rather than using non-HDL-C as a target, may be appropriate.

Our study identified the unmet need to add remnant cholesterol as a target for the secondary prevention of ASCVD. Elevated remnant cholesterol concentrations are present in patients with obesity, diabetes, and metabolic syndrome [27,28]. With the rising prevalence of cardio-metabolic disorders, the proportion of patients with ACS and elevated remnant cholesterol concentrations will continue to increase. Therefore, the atherosclerotic burden associated with remnant cholesterol is expected to rise, which poses great demands for a reduction in remnant cholesterol concentrations for the secondary prevention of ASCVD.

The major strength of our study was the use of nationwide data with a large sample size. However, some limitations should be noted. First, we calculated remnant cholesterol concentrations using fasting samples, and the cholesterol content in chylomicron remnants in the non-fasting state was not included. Therefore, the remnant cholesterol concentrations in our study population may have been underestimated. Second, the lipid profile was measured using standard hospital assays and abstracted from patients’ medical records. Although it was recommended to measure LDL-C concentrations directly in China, there might be a few of hospitals that reported the calculated LDL-C concentrations using Friedewald formula. The impact of the above situation should be limited since the clinician made the lipid-lowering treatment decisions regardless of whether or not the LDL-C level was calculated. Third, all of the patients in our study were Chinese. Therefore, the findings from our study may not be generalizable to other populations.

## 5. Conclusions

Remnant cholesterol concentrations vary among patients who are hospitalized for ACS, and one fifth have elevated remnant cholesterol concentrations at admission. Elevated remnant cholesterol concentrations are not uncommon in patients with LDL-C and/or non-HDL-C concentrations within the treatment target, which represents an unmet need to add remnant cholesterol as a target for the secondary prevention of ASCVD.

## Figures and Tables

**Figure 1 metabolites-12-00898-f001:**
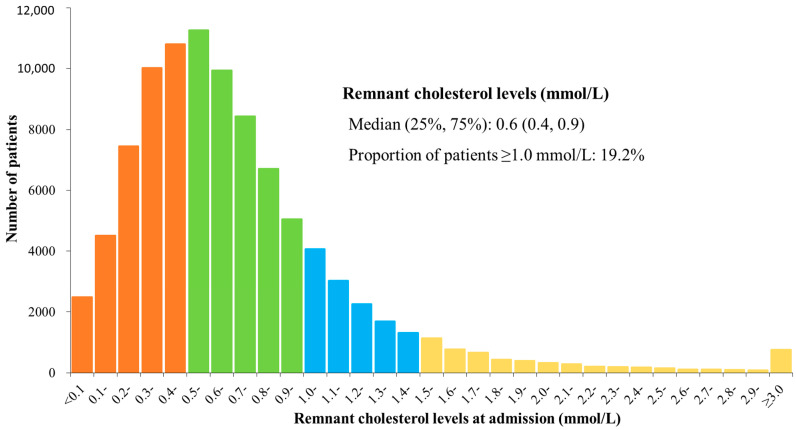
Distribution of remnant cholesterol concentrations at admission in patients with acute coronary syndrome.

**Figure 2 metabolites-12-00898-f002:**
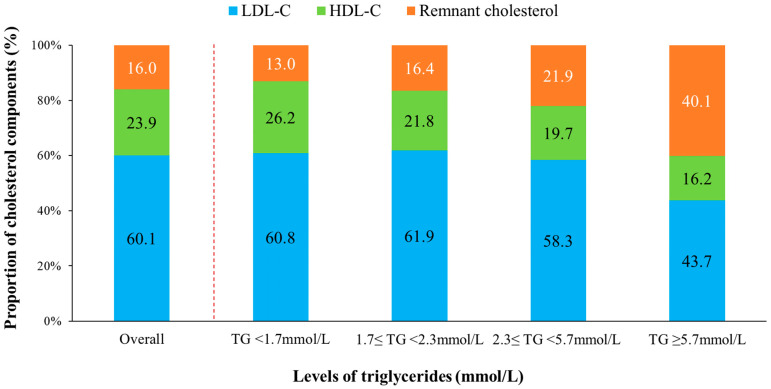
Proportions of different cholesterol components in total cholesterol among the whole patients and in subgroups with different triglyceride concentrations. LDL-C, low-density lipoprotein cholesterol; non-HDL-C, non-high-density lipoprotein cholesterol; TG, triglyceride.

**Figure 3 metabolites-12-00898-f003:**
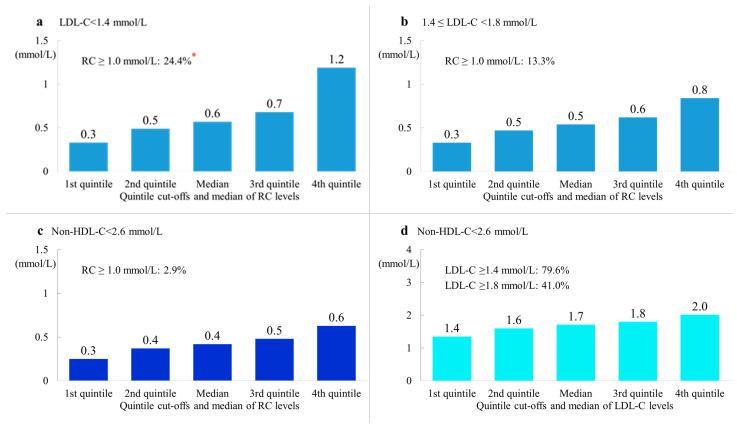
(**a**) Quintile cut-off values and the proportion of elevated remnant cholesterol concentrations in patients with LDL-C concentrations < 1.4 mmol/L. (**b**) Quintile cut-off values and the proportion of elevated remnant cholesterol concentrations in patients with LDL-C concentrations of 1.4–1.7 mmol/L. (**c**) Quintile cut-off values and the proportion of elevated remnant cholesterol concentrations in patients with non-HDL-C concentrations < 2.6 mmol/L. (**d**) Quintile cut-off values and the proportion of elevated LDL-C concentrations in patients with non-HDL-C concentrations < 2.6 mmol/L. (**e**) Quintile cut-off values and the proportion of elevated remnant cholesterol concentrations in patients with LDL-C concentrations < 1.4 mmol/L and non-HDL-C concentrations < 2.6 mmol/L. (**f**) Quintile cut-off values and the proportion of elevated remnant cholesterol concentrations in patients with LDL-C concentrations of 1.4–1.7 mmol/L and non-HDL-C concentrations < 2.6 mmol/L. *, proportion of patients with remnant cholesterol ≥ 1.0 mmol/L. LDL-C, low-density lipoprotein cholesterol; non-HDL-C, non-high-density lipoprotein cholesterol; RC, remnant cholesterol.

**Table 1 metabolites-12-00898-t001:** Characteristics of patients hospitalized for acute coronary syndrome.

Characteristics	All (n = 94,896)	Male (n = 69,674)	Female (n = 25,222)	*p* Value
	n (%) or Mean ± SD	n (%) or Mean ± SD	n (%) or Mean ± SD	
**Age, years**	63.3 ± 12.4	61.2 ± 12.4	69.0 ± 10.6	<0.001
**Type of ACS**				
STEMI	40,297 (42.5)	27,128 (38.9)	13,169 (52.2)	<0.001
NSTE-ACS	54,599 (57.5)	42,546 (61.1)	12,053 (47.8)	
**Glucose level, mmol/L**	6.9 ± 3.0	6.8 ± 2.8	7.2 ± 3.3	<0.001
**TG, mmol/L ***	1.4 (1.0, 2.0)	1.4 (1.0, 2.0)	1.5 (1.1, 2.1)	<0.001
**LDL-C at admission, mmol/L**	2.7 ± 1.0	2.7 ± 0.9	2.8 ± 1.0	
LDL-C < 1.4 mmol/L	6022 (6.4)	4560 (6.5)	1462 (5.8)	<0.001
1.4 ≤ LDL-C < 1.8 mmol/L	9312 (9.8)	7081 (10.2)	2231 (8.9)	
1.8 ≤ LDL-C < 2.6 mmol/L	30,980 (32.7)	23,316 (33.5)	7664 (30.4)	
LDL-C ≥ 2.6 mmo/L	48,582 (51.2)	34,717 (49.8)	13,865 (55.0)	
**non-HDL-C at admission (mmol/L)**	3.4 ± 1.1	3.4 ± 1.1	3.6 ± 1.2	
non-HDL-C < 2.6 mmol/L	22,116 (23.3)	16,920 (24.3)	5196 (20.6)	<0.001
2.6 ≤ non-HDL-C < 3.4 mmol/L	28,120 (29.6)	21,231 (30.5)	6889 (27.3)	
3.4 ≤ non-HDL-C < 4.1 mmo/L	21,756 (22.9)	15,752 (22.6)	6004 (23.8)	
non-HDL-C ≥ 4.1 mmo/L	22,904 (24.1)	15,771 (22.6)	7133 (28.3)	
**HDL-C at admission, mmol/L**	1.1 ± 0.3	1.0 ± 0.3	1.2 ± 0.3	<0.001
**RC at admission, mmol/L ***	0.6 (0.4, 0.9)	0.6 (0.4, 0.9)	0.6 (0.4, 0.9)	
RC ≥ 1.0 mmol/L	18,221 (19.2)	12,799 (18.4)	5422 (21.5)	<0.001
**Smoking**	38,265 (40.3)	36,301 (52.1)	1964 (7.8)	<0.001
**BMI (kg/m^2^) ^#^**	24.4 ± 3.4	24.5 ± 3.3	24.0 ± 3.6	<0.001
**Comorbidities**				
Hypertension	63,689 (67.1)	44,819 (64.3)	18,870 (74.8)	<0.001
Diabetes	41,201 (43.4)	28,746 (41.3)	12,455 (49.4)	<0.001
Prior CVD	26,011 (27.4)	18,116 (26)	7895 (31.3)	<0.001
Prior heart failure	2235 (2.4)	1285 (1.8)	950 (3.8)	<0.001
Prior chronic renal failure	1659 (1.8)	1130 (1.6)	529 (2.1)	<0.001
**Statin use in the past two weeks before admission**	16,730 (17.6)	11,882 (17.1)	4848 (19.2)	<0.001

*, Median (interquartile range) #, BMI was calculated among 59,966 patients with available information on body weight and height. BMI, body mass index; CVD, cardiovascular disease; HDL-C, high-density lipoprotein cholesterol; LDL-C, low-density lipoprotein cholesterol; non-HDL-C, non-high-density lipoprotein cholesterol; NSTE-ACS, acute coronary syndrome without ST-segment elevation; RC, remnant cholesterol; SD, standard deviation; STEMI, ST-segment elevation myocardial infarction; TG, triglyceride.

**Table 2 metabolites-12-00898-t002:** Subgroup analysis of the proportion of patients with elevated remnant cholesterol or LDL-C concentrations in those who reached different treatment targets (%).

	Proportion of Patients with Remnant Cholesterol Levels ≥ 1.0 mmol/L among Patients Reached Different Treatment Goals (%)					Proportion of Patients with LDL-C above the Target among Patients with Non-HDL-C < 2.6 mmol/L (%)	
	LDL-C < 1.4 mmol/L	1.4 ≤ LDL-C < 1.8 mmol/L	Non-HDL-C < 2.6 mmol/L	LDL-C < 1.4 + Non-HDL-C < 2.6 mmol/L	1.4 ≤ LDL-C <1.8 + Non-HDL-C < 2.6 mmol/L	LDL-C ≥ 1.4 mmol/L	LDL-C ≥ 1.8 mmol/L
**ALL**	24.4	13.3	2.9	10.9	1.2	76.9	41.0
**Type of ACS**							
STEMI	27.0	14.1	2.7	11.4	1.2	80.0	44.8
NSTE-ACS	22.4	12.4	3.2	10.5	1.2	73.8	37.2
**Prior CVD**							
Yes	17.5	11.2	2.9	8.7	1.2	72.5	35.4
No	29.5	14.5	3.0	12.6	1.2	79.6	44.4

ACS, acute coronary syndrome; CVD, cardiovascular disease; LDL-C, low-density lipoprotein cholesterol; non-HDL-C, non-high-density lipoprotein cholesterol; NSTE-ACS, acute coronary syndrome without ST-segment elevation; STEMI, ST-segment elevation myocardial infarction.

## Data Availability

The data presented in this study are available on request to the corresponding author for purposes of reproducing the results or replicating the procedure. The data are not publicly available due to privacy restrictions.

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
