# Peer review of "The Level of Remnant Cholesterol and Implications for Lipid-Lowering Strategy in Hospitalized Patients with Acute Coronary Syndrome in China: Findings from the Improving Care for Cardiovascular Disease in China—Acute Coronary Syndrome Project"

_metabolites, 2022, doi:10.3390/metabo12100898_

Round 1

Reviewer 1 Report

The authors outline the role of remnant cholesterol in Chinese patients with acute coronary syndrome. While the manuscript has several strengths, there are few issues that need to be addressed:

1. Being a study based on clinical data of the patients, maybe it will be important to specify the treatment they have/had, especially for cholesterol lowering and for how long do they take it.

2. Also, I would recommend filling the table with more data about their medical history, any familial hypercholesterolemia history they might have, number of coronary events in the last year.

Author Response

We sincerely thank you for your valuable comments and suggestions, which are truly helpful for the improvement of our manuscript. We have made corresponding changes in the manuscript according to your comments. In addition, this revised version has been edited by a professional company for English editing. All the changes can be easily recognized in the tracked version of this revised manuscript except for the English language editing, which was highlighted with red color. Our point-by-point response to each comment is listed as follows.

The authors outline the role of remnant cholesterol in Chinese patients with acute coronary syndrome. While the manuscript has several strengths, there are few issues that need to be addressed:

Point 1: Being a study based on clinical data of the patients, maybe it will be important to specify the treatment they have/had, especially for cholesterol lowering and for how long do they take it.

Response 1: We have presented the information on statins use before admission in Table 1. Statins before admission refers to statins use in the past two weeks before admission, we have made this clear in Table 1 in the revised manuscript. Unfortunately, we did not collect information regarding to the longer use of statins before admission.

Point 2: Also, I would recommend filling the table with more data about their medical history, any familial hypercholesterolemia history they might have, number of coronary events in the last year.

Response 2: We reported information on key clinical characteristics of the patients in Table 1 to keep it concise. Unfortunately, the information regarding to the familial hypercholesterolemia history and the number of coronary events in the last year were not available in our project.

Reviewer 2 Report

Dear Authors,

I congratulate you on your manuscript, in which you point out the necessity of introducing other indicators/targets for the prevention of atherogenesis. Atherosclerosis is the most common cause of death worldwide, and history has shown us that conventional therapies by lowering LDL-C and even not by raising HDL-C at all are not sufficient to successfully reverse the atherogenic process.

The introduction lacks the latest knowledge regarding the subfraction of both LDL and HDL particles in terms of their atherogenicity. Let's not forget that we know atherogenic normolipidemia as well as non-atherogenic dyslipidemia, which rests precisely on the fact that LDL-C, even if it is mostly atherogenic, is nevertheless necessary for the body, and therefore its value should be assessed in a wider context. The same, but in the opposite measure, applies to HDL fractions and to the cholesterol contained in individual HDL subfractions.

The concept of measuring cholesterol in TRLs is a very promising factor for assessing atherogenicity. The question, however, is its correct evaluation. The authors do not state the procedure for determining individual lipid markers, they state only "lipid profile was measured using standard hospital assays". In laboratories worldwide, LDL-C is not determined directly but is calculated using the Friedewald formula, so the assembly questions whether the formula "remnant cholesterol=TAG/5" (after simplification from the Friedewald formula) is correct. It is for this reason that it is questionable whether the results presented in Fig. 2 are informative: the authors divided the patients on the basis of TAG and in the subgroups they pointed out that the higher the TAG, the more remnant cholesterol, which is entirely logical, since remnant cholesterol = TAG/5... The graph and therefore the research would have a much greater informative value if the patients were not divided on the basis of TAG but on the basis of the current targets - which are mentioned both in the introduction and in the next part of the results - on the basis of whether the patients meet the LDL-C or non-HDL-C targets, or if available, based on the coronary findings.

Author Response

We sincerely thank you for your valuable comments and suggestions, which are truly helpful for the improvement of our manuscript. We have made corresponding changes in the manuscript according to your comments. In addition, this revised version has been edited by a professional company for English editing. All the changes can be easily recognized in the tracked version of this revised manuscript except for the English language editing, which was highlighted with red color. Our point-by-point response to each comment is listed as follows.

Point 1: I congratulate you on your manuscript, in which you point out the necessity of introducing other indicators/targets for the prevention of atherogenesis. Atherosclerosis is the most common cause of death worldwide, and history has shown us that conventional therapies by lowering LDL-C and even not by raising HDL-C at all are not sufficient to successfully reverse the atherogenic process. The introduction lacks the latest knowledge regarding the subfraction of both LDL and HDL particles in terms of their atherogenicity. Let's not forget that we know atherogenic normolipidemia as well as non-atherogenic dyslipidemia, which rests precisely on the fact that LDL-C, even if it is mostly atherogenic, is nevertheless necessary for the body, and therefore its value should be assessed in a wider context. The same, but in the opposite measure, applies to HDL fractions and to the cholesterol contained in individual HDL subfractions.

Response 1: Thank you for your comment. The present study tried to clarify the importance of lowering remnant cholesterol. We found that remnant cholesterol concentrations were high in patients with LDL-C and/or non-HDL-C concentrations within the target. Our findings revealed that guiding the lipid-lowering treatment only according to the LDL-C and/or non-HDL-C concentrations might not be enough. Although the latest research progress in the subfraction of LDL and HDL particles was very important to understand dyslipidemia, this information might not be pertinent to our research question.

Point 2: The concept of measuring cholesterol in TRLs is a very promising factor for assessing atherogenicity. The question, however, is its correct evaluation. The authors do not state the procedure for determining individual lipid markers, they state only "lipid profile was measured using standard hospital assays". In laboratories worldwide, LDL-C is not determined directly but is calculated using the Friedewald formula, so the assembly questions whether the formula "remnant cholesterol=TAG/5" (after simplification from the Friedewald formula) is correct. It is for this reason that it is questionable whether the results presented in Fig. 2 are informative: the authors divided the patients on the basis of TAG and in the subgroups they pointed out that the higher the TAG, the more remnant cholesterol, which is entirely logical, since remnant cholesterol = TAG/5... The graph and therefore the research would have a much greater informative value if the patients were not divided on the basis of TAG but on the basis of the current targets - which are mentioned both in the introduction and in the next part of the results - on the basis of whether the patients meet the LDL-C or non-HDL-C targets, or if available, based on the coronary findings.

Response 2: Thank you for your comment. We totally agree that the correct evaluation of remnant cholesterol concentrations is crucial for our study. The concentrations of remnant cholesterol in our study were calculated according to the formula proposed by the 2016 ESC/EAS Guidelines for management of dyslipidaemia (Eur Heart J. 2016;37(39):2999-3058).

As we have mentioned in section 2.1, our study used data from a registry study, all the information including the lipid profile was abstracted from patients’ medical records. In China, the directly measured method was recommended as the industry standard for routine measurement of LDL-C. Since all the hospitals enrolled in our study joined the lipid external quality assessment programs of the National Center for Clinical Laboratory, the quality of the measurement can be guaranteed. We have added this information in section 2.3 to make it clear (Page 3, Line 96-98).

Although it was recommended to measure LDL-C concentrations directly in China, there might be a few of hospitals that reported the calculated LDL-C concentrations using Friedewald formula. We have added this as a limitation in the limitation section (Page 10, Line 348-353).

In addition, following your suggestion, we have added the proportion of remnant cholesterol in total cholesterol according to different LDL-C and/or non-HDL-C concentrations (Page 7, Line 222-228; Supplementary Figure 1).

Reviewer 3 Report

The manuscript titled “The level of remnant cholesterol and implications for lipid 2 lowering strategy in hospitalized patients with acute coronary 3 syndrome in China: Findings from the Improving Care for Car-4 diovascular Disease in China-Acute Coronary Syndrome Pro-5 ject” is an interesting article with a large number of participants. However, some aspects needs to be elucidated.

1.- I recommed the authors to check the English style. For example, the lines 37-38 should be rephrased.

2.- Diet has a key role in caridovascular diseases, I recommed the authors to include these aspects in the introduction and discussion sections.

3.- The graphs could be improved

4.- I reccomend the authors try to associate their results wiht other variables related to cardiovascular diseases (i.e. intima-media thickness).

Author Response

We sincerely thank you for your valuable comments and suggestions, which are truly helpful for the improvement of our manuscript. We have made corresponding changes in the manuscript according to your comments. In addition, this revised version has been edited by a professional company for English editing. All the changes can be easily recognized in the tracked version of this revised manuscript except for the English language editing, which was highlighted with red color. Our point-by-point response to each comment is listed as follows.

 Point 1:  I recommend the authors to check the English style. For example, the lines 37-38 should be rephrased.

Response 1: Thank you for the suggestion. This revised version has been edited by a professional company for English editing.

Point 2: Diet has a key role in cardiovascular diseases, I recommend the authors to include these aspects in the introduction and discussion sections.

Response 2: We do agree that diet has a key role in cardiovascular diseases. Since the present study used data from a registry study, all the information were abstracted from patients’ medical records and information related to diet was not available in the medical records. Therefore, we did not include this information in our study.

Point 3: The graphs could be improved

Response 3: We have improved Figure 3 by adding notes to make it more readable. In addition, the title and notes of the figures have been edited by a company for English editing.

Point 4: I recommend the authors try to associate their results with other variables related to cardiovascular diseases (i.e. intima-media thickness).

Response 4: Thank you for your comment. Since few patients hospitalized for acute coronary syndrome received carotid ultrasonography during hospitalization, information regarding sub-clinical atherosclerosis, such as intima-media thickness, was not available in our project.

Round 2

Reviewer 3 Report

I thank the authors for considering my suggestions